# Exploring the Formation Mechanism of Unsafe Construction Behavior and Testing Efficient Occupational Health and Safety (OHS) Programs

**DOI:** 10.3390/ijerph19042090

**Published:** 2022-02-13

**Authors:** Xun Liu, Xiaobo Li

**Affiliations:** Department of Engineering Management, School of Civil Engineering, Suzhou University of Science and Technology, Suzhou 215000, China; 2013031112@post.usts.edu.cn

**Keywords:** construction, occupational health and safety (OHS), training and education, behaviorist psychology

## Abstract

Safety education and training for employees is important to ensure the safety of construction and improve the safety awareness of employees. It is difficult to meet the needs of the new situation of safety production with traditional safety education and training consequences of construction enterprises. To address this deficiency, this research analyzed the formation mechanism of safety behavior of personnel at different levels in construction enterprises from the perspective of behaviorist psychology and studied their different needs for safety training, designing training programs for simple safety behavior and complex safety behavior according to the degree of difficulty of the training content. This research also developed and tested models of training frequency and training times and carried out corresponding experimental research. Through the experiments of “safety behavior by wearing a helmet” and “fire control consciousness training”, it was found that the training of consciousness of relatively complex safety behavior should follow the principles that each training should last longer and the training interval should not be too short, so as to minimize the cost of safety education and training while ensuring the quality of safety education and training.

## 1. Introduction

The construction industry is classified as one of the most dangerous industries; workers are exposed to many accidents and risks [1,2]. The construction industry is the pillar industry of national economies [3]. Accompanied by the considerable economic contributions the construction industry makes, safety issues in the construction industry have become a major concern in many countries [4,5,6,7,8,9,10,11]. Occupational health and safety (OHS) training is a fundamental element in workplace hazard control programs [12,13]. Numerous safety and health standards for hazard control contain requirements for training aimed at reducing risk factors for injury, disease or death [14,15]. Combined with management responsibility, which is paramount, training is a necessary part of a comprehensive hazard control program [16,17]. Improving the effectiveness of OHS training efforts and other interventions is important especially as workplaces and workforces change [18,19].

To date, China is carrying out the largest infrastructure constructions in history and in the world and is also facing a severe safety situation [20]. In construction enterprises, the root cause of people’s unsafe behaviors lies in their weak awareness of safety, bad safety habits and lack of proper safety knowledge and skills [21,22]. The elimination of workers’ unsafe behaviors largely depends on the development of safety education and training [23]. Safety training for employees in Chinese construction enterprises usually refers to three-level education, on-the-job training for special operators, safety knowledge lectures, etc. [24]. Although many construction enterprises have accumulated and established a whole set of experiences and a system of safety production training, the traditional safety education training mode does not consider the formation mechanisms of unsafe behaviors at the decision level, management level and operation level of construction enterprises from the perspective of behavioral psychology. This led to its poor effectiveness, which can no longer meet the needs of the new situation of safety production; thus, the traditional training mode’s effectiveness is not ideal [25,26]. Therefore, it is necessary to innovate the means and methods of building safety education and training in order to improve safety training’s effectiveness and better serve the safety culture of construction enterprises. In order to reduce the accident rate in construction, safety trainings of construction workers must be strengthened, and safety education and training are one of the root causes of accidents [27,28]. In safety education and training, it should be noted that only when workers’ awareness, knowledge and skills have reached a certain level, and construction enterprises have achieved the transformation from material-centered to worker-oriented, can the enterprise’s safety management rise to a new stage [29,30]. Therefore, in view of the formation mechanism of unsafe construction behavior, there is an urgent need for construction enterprises to determine the safety training needs of personnel at different levels in construction enterprises and formulate scientific and reasonable training programs based on these training needs.

The present study identifies the formation mechanism of unsafe construction behavior and presents a designed framework for OHS trainings and education based on the review work on topics surrounding the formation mechanism of unsafe construction behavior. Then, we present our construction of the decision model of training frequency and training times and corresponding experimental research.

## 2. Literature Review

### 2.1. Safety Training and Safety Performance

Engineering construction has a very high risk of harm. Human factors are the most important reasons for the accidents. By conducting safety trainings, construction enterprises can increase their employees’ safety awareness, access safe operation skills and abilities to handle and respond to emergencies. This not only realizes safety in production but also highlights the need for the construction of enterprise safety culture and the importance of the rights and interests of employees [31,32,33]. The relevance of training in improving health and safety for construction workers has not only been recognized by organizations but also by workers [34]. In essence, safety training education is the training of safety culture [19,20]. The perfection degree and implementation effect of the training education system directly affect the level of safety culture of enterprises, and they are also important indicators to reflect the level of safety culture of enterprises [35].

Safety training for employees is an important measure for ensuring the safety of construction and improving the safety awareness and perceptions of employees [36,37]. Safety training is critical especially to tutor employees on safety training and compliance, which will offer prevention of accidents and controls. Safety training is also of chief concern for the success of the OHS program [38]. In addition, appropriate safety training can upgrade safety climates along with safety culture levels, which ultimately enhances effective safety performance in the construction industry [39]. A comparative study was made before and after applying safety training using Spearman’s correlation test; results showed that safety education and training can advance the level of safety climate and its relevant factors for better safety performance in construction firms [35]. Safety training is able to moderate the relationship between safety climate and safety performance in construction firms in developing countries like Nepal [38]. Safety training services are one of the mechanisms offered by legislation to promote safe behaviors of employees in construction firms [40]. Safety training acts as an adjacent latent variable that has a constructive relationship with the employer’s safety behavior [41]. Moreover, construction firms can improve workers’ protective actions through seminars, training and conferences through increasing awareness of health and safety activities [38,42]. Furthermore, the amelioration of the safety performance of construction employers is deeply affected by safety training and training technologies in any organization [20].

Designing and implementing effective safety training programs for construction workers poses complications that need to be addressed. In the present paper, we describe the design and development of an innovative safety training experiment for workers in construction. Specifically, we constructed a decision model of training frequency and training times by advancing existing training through consideration of both practical exercises and underlying theoretical learning principles.

### 2.2. Causes Affecting Safety Performance

Construction behaviors have been a major issue and their investigation has attracted the interest of researchers and practitioners. This study focused on safety behavior issues of construction workers; factors that affect construction workers’ safety performance due to policymakers, the management level of construction enterprises and operational level of construction enterprises were also discussed.

At the policymaker level, factors that may cause workers’ unsafe behaviors mainly include: excessive pursuit of economic interests [1,43,44,45], insufficient investments in safety [46,47,48], over-scheduling [49,50], unreasonable safety input and allocation of resources [18,30,51], illegal subcontracting [52,53], inadequate safety management systems [17,19,54], ignoring production safety [43,45], etc. At the management level, factors that may cause workers’ unsafe behaviors mainly include: unreasonable technical safety plans [52,55], no contingency programs [37,52], no security hardware and software protection [13,36,55,56,57], the construction plan approval not being strict [37], illegal orders [18,25,58], non-implementation of security measures [35,59], lax supervision [1,14], improper emergency handling [19,50,59], lack of vigilance [43,45], the continuation of work despite potential dangers [48,59], etc. At the operational level, factors that may cause workers’ unsafe behaviors mainly include: irregular operating [25,58], violation of construction procedures [37], carelessness [9,19,44], passive execution [22,58], a lack of sensibility towards emergencies [19,50,59], no proper security measures [35,59], excessive fatigue [50,60,61,62], etc. Unsafe behaviors of policymakers play a guiding role in safety production, leading to the occurrence of unsafe behaviors of managers and operators [58]. Considerations from the management level are mainly reflected in the safety production plan, specific measures and supervision [45], while at the operating level, physiological, psychological and technical factors have great impacts on workers’ unsafe behaviors [63]. Prior studies have identified that safety behavior has an optimistic relationship with the safety performance of a construction project [38,64,65].

## 3. Formation Mechanism of Unsafe Behaviors of Construction Workers

Perception is a reflection of human behavior; generally, the term “unsafe human behaviors” refers to those which have caused an accident or may cause an accident, including two connotations: one is behavior which causes a higher probability of an accident, and the other is behavior that is not conducive to reducing disaster loss in an accident [44,66]. There are many methods for the classification of unsafe behavior. The behavior of human insecurity can be divided into two major categories, intentional and unintentional unsafe behavior, which emphasizes the relationship between human behavior and intention [67,68]. Whether it is intentional unsafe or unintentional unsafe behavior, it can cause great harm [22,60]. Human behavior is controlled by external stimuli and brain judgment, which are then reflected by behavior. Human behavior is made up of a series of modules. Any module error may cause the final behavior to be incorrect [69,70]. About 70% of the occurrence of safety accidents are caused by unsafe human behavior. Although the formation mechanism of unsafe behavior is the same, the performance of different levels of personnel in an enterprise in the same module is different [71,72].

The unsafe behaviors of workers are caused by organizational factors or external environmental factors [73]. Unsafe behavior in engineering construction is different at different levels. The behavior of the decision-making level plays an exemplary role in the management and the operation level, seriously affecting the probability of unsafe behavior of the personnel at these two levels; that is, unsafe behavior is mostly reflected by the performance of the management and the operation levels, and it is also the most fundamental cause of safety accidents [56,74]. The management level has the greatest impact on the behavior of the operational level because its manners will affect the operational level’s behavior [43,45,58,63].

The decision-making level plays a leading role in the macro-level command and planning of enterprise development and decides the developmental direction of an enterprise. Unsafe performance of an enterprise’s decision-making level is mainly manifested in a lack of safety investments, excessively tending to economic interests when determining strategic objectives of the enterprise, the lack of safety supervision and the neglect of national laws [43]. The management plays a connecting role in the enterprise. It not only provides a basis for decision-making to the decision-making level but also distributes a concrete execution strategy to the operation level. The management’s safety behavior has a great influence on the concrete development and execution of the enterprise’s safety-producing work [45]. Improper performance of the operation level directly leads to accidents and also creates the conditions for the suffering of great physical and mental harm [50,61].

The performance of the decision-making level is the core of an enterprise’s safety production, playing an exemplary and guiding role for the management and the operation level, being responsible for the top-level design of the enterprise’s safety production and determining the input of the safety production. The performances of the decision-making, management, and operation levels are interrelated, which may eventually lead to the occurrence of security incidents. The formation mechanisms of improper performance at each level and their effects are shown in Figure 1.

## 4. Methodology

### 4.1. Content Design for OHS Trainings and Education

The worker’s cognition, judgment and coping ability surrounding construction safety risks can be improved through training and education so that they can receive relevant information for safety production correctly and make accurate judgments through thinking so as to perform the behavior of safety production [30,37]. However, the content of safety training required by personnel at different levels is different, and it is necessary to systematically identify and analyze the safety training’s objectives, knowledge structure, skill status and other aspects of personnel at different levels to determine safety training needs at different levels [62,75]. At the same time, because those at different levels of safety training have different knowledge backgrounds, the methods for their safety training should also be different [29].

The purposes of safety education and training are to improve the work performance of employees, form good safety habits of employees, reduce the probability of unsafe behavior and reduce the number of safety accidents [50]. At present, many medium- and small-sized construction enterprises in China have not set up a safety training system, or the established training system lacks practicability and is not systematic, and the training subject, training content and training method are not targeted [5,6]. The needs of personnel in safety training at different levels of construction enterprises are different; in addition, the individual quality, learning ability and learning willingness of employees are also different, so the safety training of construction enterprises should make individualized training programs for the special needs of personnel at different levels, so as to achieve the most ideal results [1,45]. To sum up, an effective safety training program is mainly reflected in three aspects: one is the pertinence of the training content, the other is the effectiveness of the training method, and the third is the rationality of the (stimulation) frequency and frequency of the training. The differences in the safety training needs of different levels of personnel in construction enterprises are shown in Table 1.

In order to make the content of safety training coincide with the different needs of construction enterprises and improve the pertinence of the training content, first of all, it is necessary to decompose and refine the training needs, and a work breakdown structure (WBS) can be used to clarify the needs of the target of the tree-layered decomposition to obtain the preliminary plan of safety training content. Secondly, it is useful to evaluate and adjust the preliminary plan by using the expert investigation method; thirdly, it is important to conduct trial training; fourthly, it is useful to evaluate and summarize the training’s effect in time, so as to further adjust the training content. This flow is shown in Figure 2.

After several rounds of cycles by participants at the decision-making level from enterprises, managers, governors, law makers and experts, relatively stable training contents were set, shown in Table 2.

### 4.2. Design of Training Frequency and Number of Training for Safety Behavior

#### 4.2.1. Simple Safety Behavior

For the training of simpler safety practices, the general rule is the process of “supervision and execution–not supervision and execution–and re-supervision and execution” [76], as shown in Figure 3.

Figure 3 shows that $t_{n1}$ is the duration of the continuation of the first (nth) supervision, PL is the implementation rate of safety behaviors at the end of the supervision interval (from 100% down to PL) and $t_{n2}$ is the interval from the first nth supervision to the second (n+1) supervision. It can be seen from practical experience that after the first (n) time of the ($t_{n1}$) time supervision, the time of $t_{n2}$ becomes longer and longer, that is, the implementation rate of safety behavior declines slowly and approaches 100% indefinitely, and the supervision interval is extended indefinitely, so as to achieve the purpose of simple safety behavior training, that is, the cultivation of simple, safe behavioral habits. Here, the purpose is to study how many times after supervision can develop simple, safe behavioral habits and how long the time of each supervision is.

#### 4.2.2. Complex Safety Behavior

The general rule of safety behavior training for complex knowledge is the process of learning to master–forgetting–learning to master again [76], as shown in Figure 4.

In Figure 4, $t_{n1}$ is the time of the nth safety knowledge learning, and PL is the rate at which the trainees have mastered the knowledge of complex safety behaviors at the end of the training and learning interval. $t_{n2}$ is the interval between the first time (n) training and the n+1 training. It can be seen from practical experience that after the first (n) time of the ($t_{n1}$) time supervision, the time of $t_{n2}$ becomes longer and longer; that is, the implementation rate of safety behavior declines slowly and approaches 100% indefinitely, and the supervision interval is extended indefinitely, so as to achieve the purpose of complex safety behavior training, Here, the purpose is to study after how many times of training the knowledge of complex safety behaviors can be firmly grasped and formed into habits and how long each training will take to learn.

## 5. Results and Discussion

Safety training should achieve two goals: one is the learning of safety behavior, the other is the development of the habit of safe work. However, learning safety behavior has a process of learning–forgetting–relearning, and the formation of safety habits has a process of supervision and execution–not supervision and execution–and re-supervision and execution. There is a question of training frequency and the number of trainings, that is, how often relearning (or re-supervising), and how many repetitions in total, will bring the forgetting rate (or the failure rate) to a tolerable level. The study found that relatively simple unsafe behaviors are mainly to save trouble, or because of fear of trouble, and so on; the kind of safety training suitable for this belongs to the habit-formation class, with wearing safety helmets being an example. The more complex safety behaviors need a training process of learning and mastering, and unsafe behavior occurs, mainly because of the forgetting of safety knowledge and wrong judgments caused by lack of safety knowledge, etc. This kind of safety training belongs to the knowledge=learning class, the use of fire extinguishers being an example. Therefore, the frequency and number of training should be designed for two different situations.

### 5.1. Generalized Model of Training Frequency and Number of Trainings

#### 5.1.1. Training Frequency and Number of Trainings for Simple Safety Behavior

The construction enterprise hopes to seek an appropriate training frequency and training times under the same supervision cost, so as to make the time to develop safety behavior shortest. Under the requirement of this goal, a generalization model of training frequency and training times was constructed. The assumptions for establishing the model were as follows:

**Hypothesis** **1.**
*trainees have the same knowledge background and learning ability;*


**Hypothesis** **2.**
*the security atmosphere of the trainee’s enterprise is the same;*


**Hypothesis** **3.**
*trainees have the same value orientation;*


**Hypothesis** **4.***the fund used by the enterprise for safety training (or safety supervision) is certain, which is*$C^{*}$.

Objective Function: $MinT(N)=\sum_{n=1}^{N}\left(t_{n-1,2}+t_{n,1}\right)$, $t_{0,n}=0$

Constrains: (1)
P=1, $\sum_{i=1}^{n}\left(t_{i,1}+t_{i,2}\right)<t\leq \sum_{i=1}^{n}\left(t_{i,1}+t_{i,2}\right)+t_{i+1,1}$, n=0,1,2,⋅⋅⋅,N

(2) $P=k_{i}\left(t\right)$, $P_{i}\leq k_{i}\left(t\right)\leq 1$, ${k_{i}}^{\prime }\left(t\right)<0$, $k_{i}{}^{\prime \prime }\left(t\right)<0$

∑i=1n(ti,1+ti,2)−tn,2<t≤∑i=1n(ti,1+ti,2), n=0,1,2,⋅⋅⋅,N

(3) $P=P_{i}$, $\sum_{i=1}^{n}\left(t_{i,1}+t_{i,2}\right)$, n=0,1,2,⋅⋅⋅,N

(4) $t_{N,2}\gg t_{N-1,2}$

(5) $C\left(N\right)=\beta \times \sum_{n=1}^{N}\left(t_{n,1}\right)\leq C*$

In the formula MinT(N), the objective function is to obtain the shortest sum of time (T) needed to develop simple and safe behaviors after the first (N) training.

**Constraint 1:** indicates that the implementation rate (P) of trainees’ simple safety behaviors during the supervision period is always 100%;

**Constraint 2:** indicates that in any interval of supervision, the implementation rate k(t) of trainees’ simple safety behaviors always falls within the interval $\left[p_{i},1\right]$;

**Constraint 3:** indicates that at the end of each supervision interval, that is, $t=\sum_{i=1}^{n}\left(t_{i,1}+t_{i,2}\right)$, the implementation rate of trainees’ simple safety behaviors is always $p_{i}$;

**Constraint 4:** indicates that the supervision interval after the last training ($t_{N-1,2}$) for trainees to develop simple safety behaviors is much larger than the previous supervision interval ($t_{N,2}$);

**Constraint 5:** indicates that the sum of supervision costs ($\beta \times \sum_{n=1}^{N}\left(t_{n,1}\right)$) is less than the cost of the enterprise’s occupational health and safety education and training input plan ($C^{*}$), and β represents the cost of unit supervision time.

#### 5.1.2. Training Frequency and Number of Trainings for Complex Safety Behaviors

The construction enterprise hopes to seek an appropriate training frequency and training times under the same training cost, so as to make the time to develop safety behavior shortest. Under the requirement of this goal, a generalization model of training frequency and training times was constructed. The assumptions for establishing the model were as follows:

**Hypothesis** **5.**
*trainees have the same knowledge background and learning ability;*


**Hypothesis** **6.**
*the security atmosphere of the trainees’ enterprise is the same;*


**Hypothesis** **7.**
*trainees have the same value orientation;*


**Hypothesis** **8.***the fund used by the enterprise for safety training (or safety supervision) is certain, which is*$C^{*}$.

Objective Function: $MinT(N)=\sum_{n=1}^{N}\left(t_{n-1,2}+t_{n,1}\right)$

Constrains: (1) $P=P_{i}$, $t=\sum_{i=1}^{n}\left(t_{i,1}+t_{i,2}\right)$, n=0,1,2,⋅⋅⋅,N

(2) P=1, $t=\sum_{i=1}^{n}\left(t_{i,1}+t_{i,2}\right)+t_{i+1,1}$, n=0,1,2,⋅⋅⋅,N

(3) $P=g_{n}\left(t\right)$, $p_{i}\leq g_{n}\left(t\right)\leq 1$, $g_{n}^{\prime }\left(t\right)<C$, $g_{n}^{\prime \prime }\left(t\right)<0$

∑i=1n(ti,1+ti,2)−tn,2<t≤∑i=1n(ti,1+ti,2), n=0,1,2,⋅⋅⋅,N

(4) $P=h_{n}\left(t\right)$, $p_{i}\leq h_{n}\left(t\right)\leq 1$, $h_{n}^{\prime }\left(t\right)>0$, $h_{n}^{\prime \prime }\left(t\right)<0$

$\sum_{i=1}^{n}\left(t_{i,1}+t_{i,2}\right)<t\leq \sum_{i=1}^{n}\left(t_{i,1}+t_{i,2}\right)+t_{i+1,1}$, n=0,1,2,⋅⋅⋅,N

(5) $t_{N,2}\gg t_{N-1,2}$

(6) $C\left(N\right)=\beta \times \sum_{n=1}^{N}\left(t_{n,1}\right)\leq C^{*}$

In the formula MinT(N), the objective function is to obtain the shortest sum of time required to fully master and execute the knowledge of complex safety behaviors after the N times training.

**Constraint****6:** indicates that at the end of each training interval, that is, $t=\sum_{i=1}^{n}\left(t_{i,1}+t_{i,2}\right)$, the trainees’ mastery rate of complex safety behavior knowledge is always $P_{i}$;

**Constraint****7:** indicates that at the end of each training period, that is, $t=\sum_{i=1}^{n}\left(t_{i,1}+t_{i,2}\right)+t_{i+1,1}$, the trainees’ mastery rate of complex safety behavior knowledge is always 100%;

**Constraint****8:** indicates that in any training interval, the trainees’ mastery rate ($g_{n}\left(t\right)$) of complex safety behavior knowledge always falls within the interval $\left[p_{i},1\right]$, and $g_{n}\left(t\right)$ is a monotonically decreasing concave function;

**Constraint****9:** indicates that in any training period, the trainees’ mastery rate ($h_{n}\left(t\right)$) of complex safety behavior knowledge always falls within the interval $\left[p_{i},1\right]$, and $h_{n}\left(t\right)$ is a monotonically decreasing convex function;

**Constraint****10:** indicates that trainees’ mastery rate of complex safety behavior knowledge reaches 100% and can always maintain this level. The interval after the last training ($t_{N,2}$) is much longer than the interval before ($t_{N-1,2}$);

**Constraint****11:** indicates that the training cost ($\beta \times \sum_{n=1}^{N}\left(t_{n,1}\right)$) is less than the safety training investment plan ($C^{*}$) of the enterprise, and β represents the cost of unit supervision time.

### 5.2. Practical Safety Performance Experiments

#### 5.2.1. Construction Helmet-Wearing Experiment

Five sites of a construction company in Suzhou were studied. One hundred workers were selected by the construction company as experimental subjects, were aged between 20 and 40 and were divided into 5 groups of 20 for comparison. We supposed the cost of supervision was 100¥ per day. The duration of each supervision was 1 day, 3 days, 7 days, 10 days and 15 days. During the supervision period, the supervisors used morning, afternoon and evening routine supervision combined with frequent surprise supervision, and the supervised subjects had to wear the helmets; that is, there should have been 20 helmets worn in each group of workers The experimental subjects did not know the monitoring frequency of the program. In the unsupervised period of the experiment, supervisors only patrolled without reminding and obtained the relevant records. Compared with supervision, the number of rounds was reduced. At the end of the supervision interval, the implementation rate of safety behavior was 70%; that is, the proportion of people who did not take the initiative to wear a helmet dropped to 70%. Experimental results were as follows:

In Group 1, $t_{n1}=1$ day, PL=70% and the number of experimental people was 20; the observation results are shown in Table 3.

In **Group 2**, $t_{n1}=3$ days, PL=70% and the number of experimental people was 20; the observation results are shown in Table 4.

In **Group 3**, $t_{n1}=7$ days, PL=70% and the number of experimental people was 20; the observation results are shown in Table 5.

In **Group 4**, $t_{n1}=10$ days, PL=70% and the number of experimental people was 20; the observation results are shown in Table 6.

In **Group 5**, $t_{n1}=15$ days, PL=70% and the number of experimental people was 20; the observation results are shown in Table 7.

By analyzing the data of supervision times, supervision intervals and supervision costs of the above experimental results, a trend chart of relevant experimental data for supervising the simple safety behavior of wearing a helmet can be drawn, as shown in Figure 5 and Figure 6.

From this experiment, we concluded that, from the above five groups of experimental data, in the case of each supervision time of helmet wearing for $t_{n,1}=1,3,5,7,10,15$ days, with each supervision time gradually extended, the number of supervision times required for trainees to develop the habit of consciously wearing helmets lowered; that is, the total supervision time could shorten, and the supervision cost could be lowered. Therefore, according to the regulation obtained from the experiment, the time of each supervision should be extended for the supervision of the formation of simple safety behavior habits so as to effectively accelerate the formation of simple safety behavior habits. In this way, the frequency and total supervision time will be shortened, and the cost of supervision will also be reduced.

#### 5.2.2. On-Site Construction Fire Drill Experiment

Five sites of a construction company in Suzhou, Jiangsu Province were studied. The company selected 100 workers between the ages of 20 and 40 as subjects. A hundred people were divided into five groups of twenty for comparison. We supposed that the training cost was 100¥ per person per day and chose five training programs. For the first kind, the training time of fire fighting was 7 days, and the interval time of the two trainings was selected as January. For the second kind, the training time of fire fighting was 14 days, and the interval time of the two trainings was selected as February. For the third kind of training, the training time of fire fighting was 21 days, and the interval time of the two trainings was selected as March. For the fourth kind, the training time of fire fighting was 28 days, and the interval time of the two trainings was selected as May. For the fifth kind, the duration of fire training was 35 days, and the interval time of the two trainings was selected as August.

Training methods were a combination of theory and practice. Theoretical training refers to teaching construction site fire safety knowledge courses; practical training refers to basic operational guidance for construction site fires. Among them, the knowledge points of the fire safety course mainly included: fire classification and characteristics, construction sites prone to fire, fire safety laws and regulations, fire prevention of key parts and key types of work, installation and commissioning of fire prevention, other construction and life fire preventions, common fire extinguishers and uses in the construction site and other knowledge points. The basic fire operation training mainly included: the use of foam and dry powder fire extinguishers, the understanding and use of high-pressure water pumps and the fire prevention operation of electric welding and gas cutting operators (screening the welding cutting environment and the welding object, checking fire extinguishing equipment, leaving the field, monitoring gas operation, etc.).

The time distribution of the five training programs was as follows:

When the time of the fire fighting training was 7 days, 5 days were used for theoretical training, and 2 days were used for practical operation training.

When the time of the fire fighting training was 14 days, 9 days were used for theoretical training, and 5 days were used for practical operation.

When the time of the fire fighting training was 21 days, 14 days were used for theoretical training, and 7 days were used for practical operation.

When the firefighting training time was 28 days, 15 days were used for theoretical training, and 13 days were used for practical operation.

When the firefighting training time was 35 days, 17 days were used for theoretical training, and 18 days were used for practical operation.

The longer the training time of each training program, the larger the $T_{\mathrm{Theory}}/T_{\mathrm{Practice}}$. The design is suitable for the scientific allocation of theoretical and practical time in “sports” and other training.

The assessment used the examination paper assessment method and included all the key knowledge points in the theoretical training and practical operation training. The paper design adopted the form of a question bank that contains all the examination knowledge. In each examination period, a number of questions were randomly selected for each group of subjects to answer. According to the score of each person (full score 1, all right 1, partial correct percentage score), the score of 20 people was summed, and the score of each group was averaged.

Note: While this experiment did not arrange the large fire safety drill, the examination of the content of the exercise and the drill operation was still reflected in the test paper questions and answers. This experiment did not set the zero boundary point constraint of the safety behavior implication rate; the purpose of this experiment was to accurately simulate the regulation of the experimental object to master complex safety knowledge. For the different training programs, the assessment frequency was also different; according to training time intervals of 2 days, 4 days, 6 days, 10 days and 16 days, five kinds of assessment frequency to make the relevant records were used, and the assessment frequency could be adjusted flexibly according to the actual situation of the construction site.

Experimental results were as follows:

In Group 1, $t_{n1}=7$ days, $t_{n2}=1$ month and the number of experimental people was 20; the observation results are shown in Table 8.

In Group 2, $t_{n1}=14$ days, $t_{n2}=2$ months and the number of experimental people was 20; the observation results are shown in Table 9.

In Group 3, $t_{n1}=21$ days, $t_{n2}=3$ months and the number of experimental people was 20; the observation results are shown in Table 10.

In Group 4, $t_{n1}=28$ days, $t_{n2}=5$ months and the number of experimental people was 20; the observation results are shown in Table 11.

In Group 5, $t_{n1}=35$ days, $t_{n2}=8$ months and the number of experimental people was 20; the observation results are shown in Table 12.

By analyzing the data of training times, training intervals and training costs of the above experimental results, a trend chart of relevant experimental data for training the complex safety behavior for fire drills on construction sites can be drawn, as shown in Figure 7 and Figure 8.

From the above five groups of experimental data, it can be seen that the third group had the lowest training time and training cost, that is, $t_{n1}=21$ days and $t_{n2}=3$ months, and the minimum training amount under the premise of meeting the requirements was two. Based on Figure 7 and Figure 8, the training cost of the fifth group was obviously higher than that of other groups, and the interval training of the fifth group was 8 months; therefore, although each training lasted longer (35 days), a long interval (August) likely caused the forgetfulness of the knowledge of the workers of complex safety behavior, so the complex safety behavior education training mechanism design should try to avoid the phenomenon of long training intervals. At the same time, in the case of short training intervals, such as days and months, it is not advisable to require frequent training due to the short duration of each training and the low level of mastery of complex safety behavior knowledge, thus increasing the training times and related training costs. In conclusion, the design of the complex safety behavior training mechanism should follow the principle that each training lasts a long time and the training interval should not be too short, so as to minimize the cost of safety education and training on the basis of ensuring the quality of safety education and training.

## 6. Conclusions

From the perspective of behavioral psychology, this paper analyzes the formation mechanism of unsafe behaviors at the decision-making, management and operation levels of construction enterprises and concludes that unsafe behaviors at the decision, management and operation levels are multi-interrelated. Firstly, based on the content of safety trainings for personnel of different levels in construction enterprises, it is concluded that decision-makers should mainly focus on education in the aspects of hazard sources, safety benefits, cognition of laws and regulations, safety consciousness, and safety morale levels. The management should focus on safety management technology (the correct formulation and implementation of regulations), risk management, accident emergency and other aspects of education. The operation level needs to train in safety knowledge, safety skills knowledge, safety attitudes, safety laws and regulations and other aspects of training. Further, according to the difficulty of training content, simple safety behaviors and complex safety behaviors were separated into two training programs, the generalized model of training frequency and training frequency was constructed and the corresponding experimental research was carried out. Through the safety behavior and training experiments, it was found that the training of consciousness of relatively complex safety behavior should follow the principle that each training should last longer and the training interval should not be too short, so as to minimize the cost of safety education and training on the basis of ensuring the quality of safety education and training.

## Figures and Tables

**Figure 1 ijerph-19-02090-f001:**
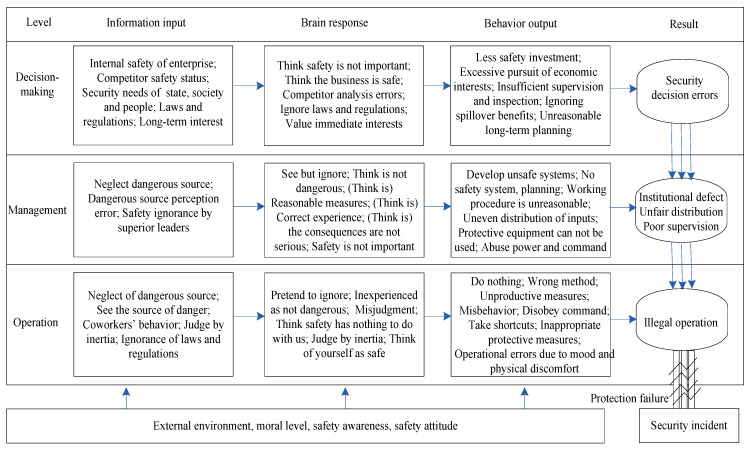
Formation mechanism and influence of improper performance at each level.

**Figure 2 ijerph-19-02090-f002:**
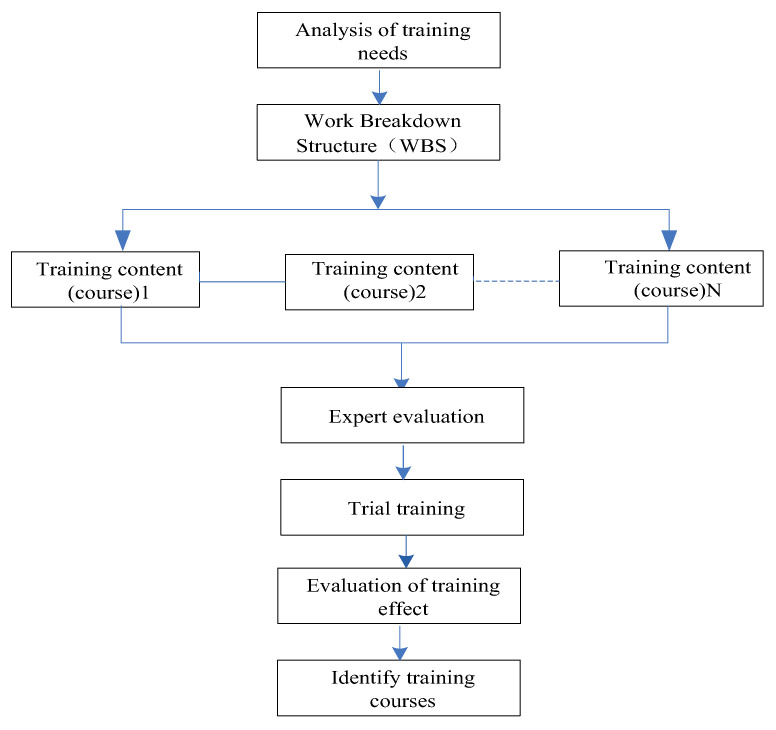
Training content confirmation process.

**Figure 3 ijerph-19-02090-f003:**
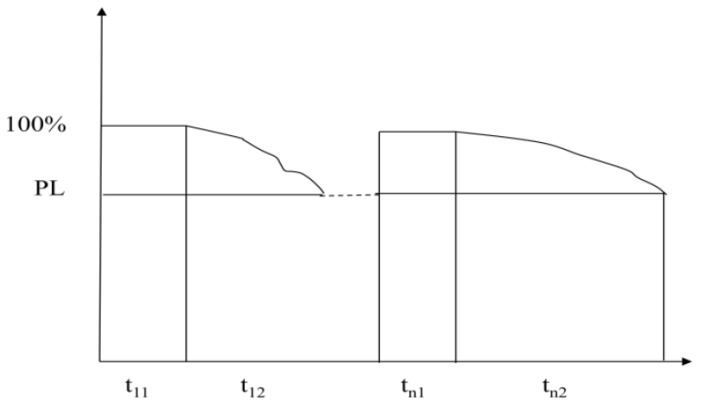
Concept of training frequency and number of training times for simple safety behavior.

**Figure 4 ijerph-19-02090-f004:**
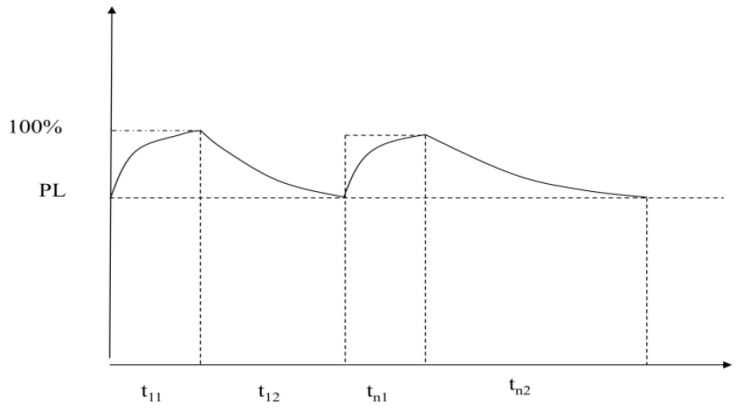
Concept of training frequency and number of training times for complex safety behavior.

**Figure 5 ijerph-19-02090-f005:**
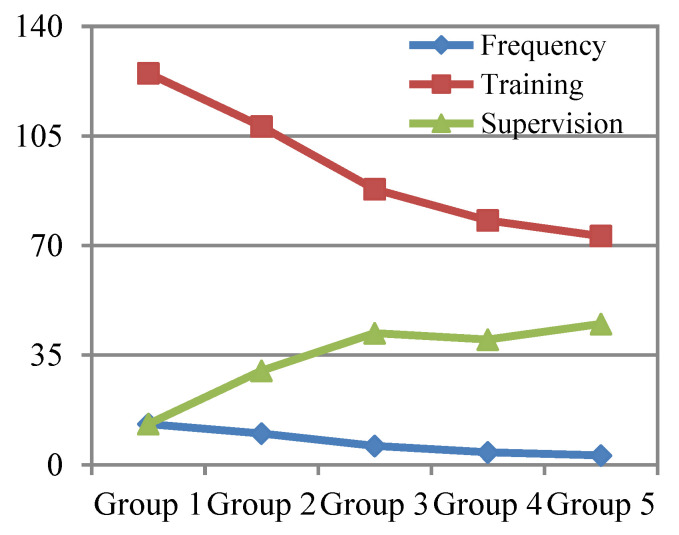
Comparison of experimental data and trends of simple safety behavior in each group.

**Figure 6 ijerph-19-02090-f006:**
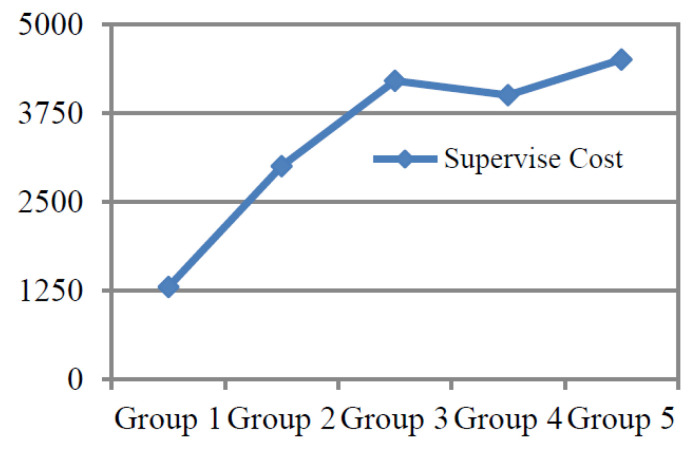
Supervision cost curve of each group experiment for simple safety behavior.

**Figure 7 ijerph-19-02090-f007:**
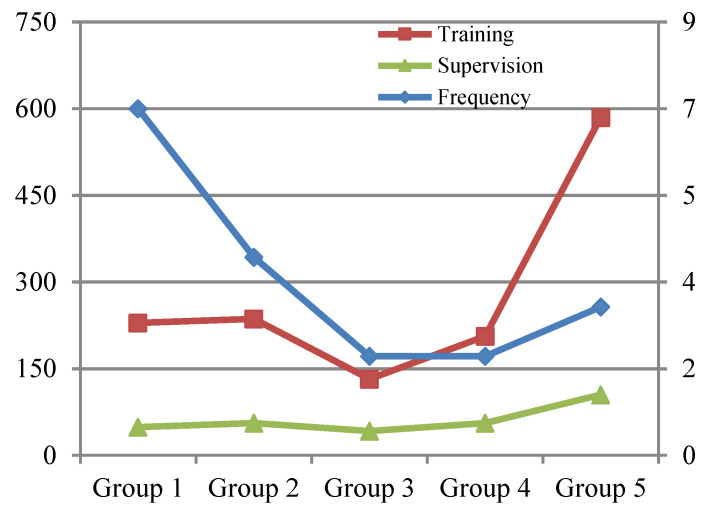
Comparison of relevant data and trends of each group of experiments on complex safety behaviors.

**Figure 8 ijerph-19-02090-f008:**
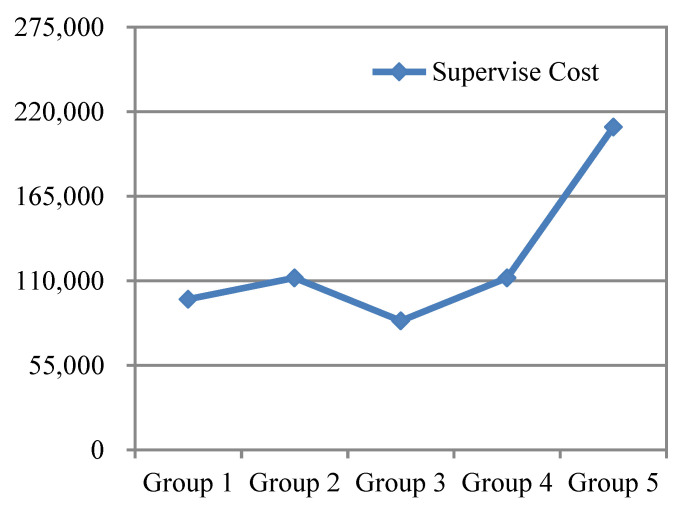
Supervision cost curve of each group of experiments for complex safety behaviors.

**Table 1 ijerph-19-02090-t001:** Training objectives at all levels of construction enterprises.

Level	Training Requirements (Objectives)
Decision-making level	The philosophy of safety first, the economic outlook of safety as beneficial, the emotional outlook of respect for life and the scientific outlook of prevention
Management level	Safe management responsibilities, safety management skills and emergency management capabilities
Operation level	The attachment of great importance to safety, ensuring the safety operation conforms to the standard requirements and actively implementing the safety production

**Table 2 ijerph-19-02090-t002:** Object, purpose and contents of OHS trainings and education.

Object	Purpose	Contents
Enterprise decision-making level, such as legal representatives, party and government leaders, directors at all levels	Create the philosophy of safety first Create the emotional outlook of respect for life Create the economic outlook of safety being beneficial Instill the scientific outlook of prevention	Work safety guidelines, policies, laws and regulations Safety production management capability Correct security thinking Realistic work style Advanced safety production management experience Typical accident case analysis
Enterprise management, such as middle managers and grass-roots managers, team leaders	Increase awareness and senses of responsibility Increase care and attach importance to safe production Set a good example Support safety work	Work safety guidelines, policies, laws and regulations Employee safety production responsibility system In-depth analysis of typical accident cases Basic safety technical knowledge Policies, laws and regulations on work-related injury insurance Statistics, reports, investigation and treatment of casualties and occupational diseases On-site inspection techniques and temporary emergency measures Management of major hazard sources and preparation of emergency rescue plans Advanced safety production management experience
New employees (including contract workers), temporary workers, trainees, interns and students, etc.	Training and mastering of the basic skills required for the position	Three-level safety education **On-site safety education** Basic knowledge of production safety; rules and regulations of production safety of the unit; labor safety discipline; dangerous factors, preventive measures and emergency plans in production sites and public posts; accident cases, etc. **Construction safety education** The safety production status and rules and regulations of the section, district and team; dangerous factors, preventive measures and emergency measures existing in the production site and working positions; typical accident case analysis **Team safety education** Job safety operation rules; proper use of production equipment, safety devices and labor protection articles (appliances); specific accident cases, etc.
Special operations personnel for operations such as electrical work, metal welding cutting, lifting machinery, erection operations, refrigeration, blasting operations	Improve professional skills, hold the certification	Theoretical knowledge Actual operation Note: according to relevant national standards

**Table 3 ijerph-19-02090-t003:** Construction Helmet-Wearing Experimental observation results of the first group.

Days	1	2	3	4	5	6	7	8	9	10	11	12	13	14	15	16	17
No.1	16	16	14														
No.2	16	17	16	14													
No.3	17	17	16	15	14												
No.4	18	17	17	17	16	15	14										
No.5	18	18	17	18	17	17	15	14									
No.6	19	19	19	19	19	18	16	16	16	14							
No.7	19	19	19	18	18	19	19	19	15	16	14						
No.8	20	19	20	19	19	19	17	18	18	17	16	14					
No.9	19	20	20	20	20	19	19	18	16	16	16	14					
No.10	20	20	20	20	18	18	17	17	17	18	16	16	14				
No.11	20	20	19	20	18	18	19	19	16	16	17	17	14				
No.12	20	20	20	19	19	19	19	18	18	15	18	15	15	14			
No.13	20	20	20	20	20	19	19	20	20	18	19	19	20	19	18	19	20

Relevant experimental data: n=3, $\sum_{n=1}^{N}\left(t_{n,1}\right)=n*t_{n,1}=13$, $C=\beta \times \sum_{n=1}^{N}\left(t_{n,1}\right)=1300$, T=125.

**Table 4 ijerph-19-02090-t004:** Construction Helmet-Wearing Experimental observation results of the second group.

Days	1	2	3	4	5	6	7	8	9	10	11	12	13	14	15	16	17
No.1	18	18	16	14													
No.2	18	19	18	16	14												
No.3	19	18	18	18	15	14											
No.4	19	19	18	17	18	18	14										
No.5	19	19	19	17	17	16	15	15	14								
No.6	20	20	20	18	19	19	18	17	17	14							
No.7	20	20	20	20	20	20	18	18	16	16	14						
No.8	20	20	20	20	19	20	17	17	17	15	15	14					
No.9	20	19	20	18	18	20	20	18	18	16	16	15	15	14			
No.10	20	19	20	18	19	20	20	18	18	18	18	18	19	18	19	18	18

Relevant experimental data: n=10, $\sum_{n=1}^{N}\left(t_{n,1}\right)=n*t_{n,1}=30$, $C=\beta \times \sum_{n=1}^{N}\left(t_{n,1}\right)=3000$, T=108.

**Table 5 ijerph-19-02090-t005:** Construction Helmet-Wearing Experimental observation results of the third group.

Days	1	2	3	4	5	6	7	8	9	10	11	12	13	14	15	16	17
No.1	20	20	19	19	17	14											
No.2	20	20	20	17	18	17	14										
No.3	20	20	19	19	18	18	16	14									
No.4	20	20	20	19	18	18	18	16	15	14							
No.5	20	20	20	19	18	18	18	19	19	18	16	16	14				
No.6	20	20	18	19	19	20	20	19	19	19	20	19	20	17	18	19	18

Relevant experimental data: n=6, $\sum_{n=1}^{N}\left(t_{n,1}\right)=n*t_{n,1}=42$, $C=\beta \times \sum_{n=1}^{N}\left(t_{n,1}\right)=4200$, T=88.

**Table 6 ijerph-19-02090-t006:** Construction Helmet-Wearing Experimental observation results of the fourth group.

Days	1	2	3	4	5	6	7	8	9	10	11	12	13	14	15	16	17
No.1	20	20	19	19	19	20	18	14									
No.2	20	20	20	20	19	19	16	16	16	14							
No.3	20	20	20	20	19	19	18	18	15	16	14						
No.4	20	18	19	20	20	20	19	19	20	20	17	18	20	18	19	20	19

Relevant experimental data: n=4, $\sum_{n=1}^{N}\left(t_{n,1}\right)=n*t_{n,1}=40$, $C=\beta \times \sum_{n=1}^{N}\left(t_{n,1}\right)=4000$, T=78.

**Table 7 ijerph-19-02090-t007:** Construction Helmet-Wearing Experimental observation results of the fifth group.

Days	1	3	5	7	8	9	10	11	12	13	14	15	16	17	18	19	20
No.1	20	20	20	20	20	20	20	20	18	16	14						
No.2	20	20	20	20	20	19	19	19	18	18	14						
No.3	20	20	20	19	19	18	18	19	19	19	20	20	19	19	20	20	18

Relevant experimental data: n=3, $\sum_{n=1}^{N}\left(t_{n,1}\right)=n*t_{n,1}=45$, $C=\beta \times \sum_{n=1}^{N}\left(t_{n,1}\right)=4500$, T=73.

**Table 8 ijerph-19-02090-t008:** On-Site Construction Fire Drill Experimental observation results of the first group.

Days	1	3	5	7	9	11	13	15	17	19	21	23	25	27	29	30
No.1	18	18	17	15	16	17	15	14	14	14	13	13	12	12	11	12
No.2	17	18	16	17	16	15	17	16	16	15	15	15	14	14	13	13
No.3	19	18	19	18	19	18	17	16	18	16	17	16	16	16	15	15
No.4	20	19	19	18	18	17	17	17	17	16	16	16	17	15	15	15
No.5	19	19	20	19	18	19	17	18	18	18	18	17	17	17	17	16
No.6	20	20	20	20	19	19	19	19	18	17	18	18	17	18	17	17
No.7	20	20	19	20	19	19	20	19	20	19	20	20	20	19	19	20

Relevant experimental data: n=7, $\sum_{n=1}^{N}\left(t_{n,1}\right)=n*t_{n,1}=49$, $C=\beta \times \sum_{n=1}^{N}\left(t_{n,1}\right)=98,000$, T=229.

**Table 9 ijerph-19-02090-t009:** On-Site Construction Fire Drill Experimental observation results of the second group.

Days	2	6	10	14	18	22	26	30	34	38	42	46	50	54	58	60
No.1	17	18	16	17	16	15	17	16	17	17	16	16	16	16	15	14
No.2	19	18	19	18	19	18	17	16	18	17	16	16	16	17	16	15
No.3	20	20	19	20	18	19	17	17	18	16	17	18	17	18	17	18
No.4	20	19	20	18	19	20	18	19	20	20	19	18	20	20	20	19

Relevant experimental data: n=4, $\sum_{n=1}^{N}\left(t_{n,1}\right)=n*t_{n,1}=56$, $C=\beta \times \sum_{n=1}^{N}\left(t_{n,1}\right)=112,000$, T=236.

**Table 10 ijerph-19-02090-t010:** On-Site Construction Fire Drill Experimental observation results of the third group.

Days	3	9	15	21	27	33	39	45	51	57	63	69	75	81	87	90
No.1	20	19	19	19	19	19	19	18	18	18	18	18	18	16	17	17
No.2	20	20	19	20	19	20	19	20	20	19	20	20	19	20	19	20

Relevant experimental data: n=2, $\sum_{n=1}^{N}\left(t_{n,1}\right)=n*t_{n,1}=42$, $C=\beta \times \sum_{n=1}^{N}\left(t_{n,1}\right)=84,000$, T=132.

**Table 11 ijerph-19-02090-t011:** On-Site Construction Fire Drill Experimental observation results of the fourth group.

Days	5	15	25	35	45	55	65	75	85	95	105	115	125	135	145	150
No.1	20	20	20	20	20	19	19	19	18	19	18	17	16	17	16	17
No.2	20	20	20	20	20	20	20	20	19	20	20	19	20	19	19	20

Relevant experimental data: n=2, $\sum_{n=1}^{N}\left(t_{n,1}\right)=n*t_{n,1}=56$, $C=\beta \times \sum_{n=1}^{N}\left(t_{n,1}\right)=112,000$, T=206.

**Table 12 ijerph-19-02090-t012:** On-Site Construction Fire Drill Experimental observation results of the fifth group.

Days	8	24	40	56	72	88	104	120	136	152	160	184	200	216	232	240
No.1	20	20	20	20	20	20	20	20	19	19	19	19	18	18	17	17
No.2	20	20	19	20	19	19	20	20	19	18	18	18	18	18	17	16
No.3	20	20	20	20	19	20	20	20	19	20	19	19	20	20	20	19

Relevant experimental data: n=3, $\sum_{n=1}^{N}\left(t_{n,1}\right)=n*t_{n,1}=105$, $C=\beta \times \sum_{n=1}^{N}\left(t_{n,1}\right)=210,000$, T=585.

## Data Availability

The data presented in this study are available on request from the corresponding authors.

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
