# Peer review of "Exploring the Formation Mechanism of Unsafe Construction Behavior and Testing Efficient Occupational Health and Safety (OHS) Programs"

_ijerph, 2022, doi:10.3390/ijerph19042090_

Round 1

Reviewer 1 Report

The article has been sufficiently improved. I look forward to reading the article in the journal.

Author Response

We appreciate the time and effort of the Reviewers and the Editor in reviewing our manuscript. The reviews are very helpful for us to improve the manuscript. As a result of the comments from both the Editor and the Reviewers we have made significant changes and have rewritten parts of the manuscript. Point to point respond to all comments are as follows. Revised parts are marked in RED color in the revised version.

Reviewer 2 Report

The article presents a very interesting and topical subject. It should be noted that it is, however, preliminary research. It will be necessary in the future to expand research and increase the research sample. Pay attention to the rich and very up-to-date bibliography.

Author Response

Response to the Editor and Reviewers’ Comments

Manuscript ID: ijerph-1543685

Title: “Exploring the Perceptions of Construction Workers for Occupational Health and Safety (OHS)Training Program through the Formation Mechanism of Construction Unsafe Behavior”

Author(s): Xun Liu, Xiaobo Li

We appreciate the time and effort of the Reviewers and the Editor in reviewing our manuscript. The reviews are very helpful for us to improve the manuscript. As a result of the comments from both the Editor and the Reviewers we have made significant changes and have rewritten parts of the manuscript. Point to point respond to all comments are as follows. Revised parts are marked in RED color in the revised version.

Reviewer 2:

Comment:

The article presents a very interesting and topical subject. It should be noted that it is, however, preliminary research. It will be necessary in the future to expand research and increase the research sample. Pay attention to the rich and very up-to-date bibliography.

Response:

We thank the reviewer’s comments, we have reorganized the whole structure of our manuscript and have updated references. The added references were marked in red, please refer to the revised version.

Reviewer 3 Report

This research analyzed the formation mechanism of safety behavior of personnel at different levels from the perspective of behaviorist psychology, and studied their different needs for safety training, designs the training program of simple safety behavior and complex safety behavior according to the degree of difficulty of training content. This research also developed and tested models of training frequency and training times, and carried out corresponding experimental research. This article discussed two interesting topics, both of which were valuable in the context of construction safety study. However, the organization of the paper should be modified, and concepts and terms used in the article are unclear. To sum up, more work should be done before reconsidering this article as a qualified one for publishing.

  1. Most of the concepts, definition and relevant theories of “unsafe behavior” used in the article, especially for the literature review section, mainly refers to the operation level. On the other hand, as for the decisions made by policymakers or management level that affect the safety performance, these behaviors are usually not attributed to scope of the term “unsafe behavior”. If the authors expect to highlight the differences in impacts posed by different levels of personnel relating to construction safety, the terms and concepts should be clarified. Accordingly, the methods, results and discussion concerning to the works belonging to section 3.1 and section 3.2 should be more specifically described.

  1. As for the experimental section, the article conducted two experiments at the operation level to identify the influence of frequency and time in the safety training program. However, it is notable that the content in this section almost detached from the literature review section in the articles. Namely, the core issues proposed in the literature review section had not been answered in the subsequent sections in this article.

  1. Similarly, the confusing terms and insufficient explanation applied in the article can be also seen in introduction to the supervision cost and time/times. This section and corresponding trial and validation experiment design should be clarified and elaborated if authors determined to emphasize this part.

  1. A few contents in the Methodology section should be reorganized as results, or introduced with more appropriate ways instead of figures.

Author Response

Response to the Editor and Reviewers’ Comments

Manuscript ID: ijerph-1543685

Title: “Exploring the Perceptions of Construction Workers for Occupational Health and Safety (OHS)Training Program through the Formation Mechanism of Construction Unsafe Behavior”

Author(s): Xun Liu, Xiaobo Li

We appreciate the time and effort of the Reviewers and the Editor in reviewing our manuscript. The reviews are very helpful for us to improve the manuscript. As a result of the comments from both the Editor and the Reviewers we have made significant changes and have rewritten parts of the manuscript. Point to point respond to all comments are as follows. Revised parts are marked in RED color in the revised version.

Reviewer 3:

This research analyzed the formation mechanism of safety behavior of personnel at different levels from the perspective of behaviorist psychology, and studied their different needs for safety training, designs the training program of simple safety behavior and complex safety behavior according to the degree of difficulty of training content. This research also developed and tested models of training frequency and training times, and carried out corresponding experimental research. This article discussed two interesting topics, both of which were valuable in the context of construction safety study. However, the organization of the paper should be modified, and concepts and terms used in the article are unclear. To sum up, more work should be done before reconsidering this article as a qualified one for publishing.

Comment 1:

Most of the concepts, definition and relevant theories of “unsafe behavior” used in the article, especially for the literature review section, mainly refers to the operation level. On the other hand, as for the decisions made by policymakers or management level that affect the safety performance, these behaviors are usually not attributed to scope of the term “unsafe behavior”. If the authors expect to highlight the differences in impacts posed by different levels of personnel relating to construction safety, the terms and concepts should be clarified. Accordingly, the methods, results and discussion concerning to the works belonging to section 3.1 and section 3.2 should be more specifically described.

Response:

We thank for the reviewer’s comments and suggestions. Firstly, we have reorganized the structure of the manuscript. Previous section 3.1 was taken separately as “section 3 Formation mechanism of unsafe behaviors of construction workers”, and previous “3.2 Content design for OHS safety trainings and education” was moved to “4 Methods 4.1. 4.1Content design for OHS safety trainings and education”

Also, We have made revisions as for the term clarifications, please refer to the contents in Literature Review in the revised manuscript. In literature review and section 3&4, we have clarified the differences between unsafe behavior of workers and improper performance from decision making level, management level and operation level. The revised parts have been marked in red color.

Comment 2:

As for the experimental section, the article conducted two experiments at the operation level to identify the influence of frequency and time in the safety training program. However, it is notable that the content in this section almost detached from the literature review section in the articles. Namely, the core issues proposed in the literature review section had not been answered in the subsequent sections in this article.

Response:

We thank for the reviewer’s comments. As for the two conducted safety performance experiments, the following contents were added in in literature review section:

“2.1 Safety training and safety performance

Engineering construction has a very high risk of harm, human factor is the important reason for the accident, by conducting safety training, can make the employee safety awareness, access to safe operation skills, handle and respond to an emergency ability, this is not only realize the safety in production, the need of the rights and interests of employees, but also the need of the construction of enterprise safety culture[31-33]. The relevance of training in improving health and safety for construction workers has not only been recognized by organizations, but also by workers[34].In essence, safety training education is the training of safety culture[19, 20]. The perfection degree and implementation effect of the training education system directly affect the level of safety culture of enterprises, and it is also an important indicator to reflect the level of safety culture of enterprises[35].

Safety training for employees is an important measure to ensure the safety of construction and improve the safety awareness and perception of employees[36, 37]. Safety training is critical, specially, to tutor employees on safety training and compliance, which will offer prevention of accidents and controls. Besides, safety training is also chief concerned for the successful of OSH program[38]. In addition, appropriate safety training can upgrade safety climate along with safety culture level which ultimately enhances effective safety performance in construction industry[39]. A comparative study was made before and after applying safety training using Spearman's correlations test, result showed that safety education and training can advance the level of safety climate and its relevant factors for better safety performance in construction firm[35]. Safety training is able to moderate the relationship between safety climate and safety performance in construction firm in developing country like Nepal[38]. Safety training services are one of the mechanisms offered by legislation to promote safe behaviors of employees in construction firm[40]. Safety training acts as an adjacent latent variable that has a constructive relationship with the employer’s Safety behavior[41]. Moreover, construction firms can improve workers' protective actions through seminars, training and conferences through increasing awareness of health and safety activities[38, 42]. Furthermore, the ameliorate of safety performance of construction employers deeply lean back in safety training and training technologies in any organization[20].

Designing and implementing effective safety training programs for construction workers poses complications that need to be addressed. In the present paper, we described the design and development of an innovative safety training experiment for workers in construction. Specifically, we constructed decision model of training frequency and training times by advancing existing training through consideration of both practical exercises and underlying theoretical learning principles.”

Comment 3:

Similarly, the confusing terms and insufficient explanation applied in the article can be also seen in introduction to the supervision cost and time/times. This section and corresponding trial and validation experiment design should be clarified and elaborated if authors determined to emphasize this part.

Response:

We thank for the reviewer’s comments. In the present study, we have made hypothesis that the fund used by enterprise for safety supervision cost is certain which is expressed by C*, and used unit supervision time β, the supervision cost and time at different levels were not emphasized. 

Comment 4:

A few contents in the Methodology section should be reorganized as results, or introduced with more appropriate ways instead of figures.

Response:

We thank for the reviewer’s suggestions, we have moved contents of “generalized model” from section methodology to Results and Discussion. Please refer to 5.1.

This manuscript is a resubmission of an earlier submission. The following is a list of the peer review reports and author responses from that submission.

Round 1

Reviewer 1 Report

Dear Authors,

The abstract needs to be summarized as it longer than it is expected and especially

"Through the experiment of "how to develop the safety behavior of wearing a helmet", it was found that when the duration of supervision was 10 days and the duration of supervision was 4 times, the optimal duration of safety behavior was 78 days. Through the experiment of "how to master fire control knowledge and cultivate good fire control consciousness", it was found that the training of consciousness of relatively complex safety behavior should follow the principle that each training should last longer and the training interval should not be too short, so as to minimize the cost of safety education and training on the basis of ensuring the quality of safety education and training"

I really appreciate your valuable paper

kind regards,

Author Response

(The authors gave the same response as above.)

Reviewer 2 Report

The topic of this study is within the scope the journal. The following are a few suggestions for improvement.

  1. In the “Introduction” section, it mentioned “the training effect is not ideal”. There is a need of more exploration about this so that readers can have a better understanding about the research background.
  2. In the “Introduction” section, it mentioned “The traditional safety education and training effect is poor, and it can no longer meet the needs of the new situation of safety production”. There is a need of more exploration about this, such as what the meaning of “the new situation of safety production”?
  3. According to the title, the study focused on the “perceptions of construction workers for OHS training program”. Nevertheless, the Introduction section does not include sufficient background information related to OHS perceptions of workers.
  4. In the Introductions section, the research gap and aim are not very clear.
  5. There is a need of a literature review section. For instance, some historical literature contents in the sections 1 and 2 can be combined to be a literature review section.
  6. The research methodology section includes various historical literature discussions, which should be moved to a literature review section. In addition, the descriptions about the research methods and process used in this study are very short. A detailed explanation about the research methods and process used in this study should be provided. For instance, it is not very clear how the figure 2 was developed and how the table 2 were derived. Moreover, based on the title, the study conducted the research through formation mechanism. Nevertheless, there are no explanations about it (and how it was used) in the research method part.
  7. It is suggested to double-check the structure of the study. The section 3 “Results and Discussion” includes a lot of contents that are related to methods. In-depth discussions about the findings of this study are very limited in this section.
  8. Both the theoretical and practical implications of this study should be more discussed.

Author Response

(The authors gave the same response as above.)

Reviewer 3 Report

The title and abstract are misleading. this is a great study examining the effect of training and education on safety behaviour, workers perception and cost-benefit analysis of training frequency and training cost versus increased supervision costs.

The abstract needs rewriting with the actual claim of the research unit and duration. It is not clear what the claim is and how this relates to training and education. A more explicit terminology would be welcome.

Terminology needs attention. Safety production or performance?

After the identification of the research gap and justification of why this study is needed, add the research aim or purpose and how this will be done/research approach, before the structure of the rest paper.

This is a welcome study that needs to be amended. Structurally I would recommend moving the literature review and research framework before the research methods section (not a methodology as it stands). The claims from the results do not do the collected evidence justice and the discussion needs to refer back to the literature and compare and synthesise the results. Conclusions are currently misleading and need to be updated in accordance with results and findings from the discussion with the literature. also, relate back to the research questions.

Author Response

(The authors gave the same response as above.)

Round 2

Reviewer 2 Report

Thanks for the revision of this paper. A few suggestions for improvement:

  1. In the response document, the reviewer’s comments were combined, and the revised parts are highlighted in the revised paper (for most of these comments, there are no specific explanations about how these comments are addressed). Nevertheless, after carefully reading the revised parts, some of the previous comments are not well addressed. For instance, there is no detailed explanation about “The traditional safety education and training effect is poor, and it can no longer meet the needs of the new situation of safety production” in the revised paper. What is the meaning of “the new situation of safety production”?
  2. It is suggested that the authors respond to reviewers’ comments one by one, which will make the response clearer.
  3. The “formation mechanism” is an important concept of this study. Nevertheless, there are no explanations about the meaning of “formation mechanism” in the “Introduction” section.
  4. According to the title, it seems that the study will explore “the perceptions of construction workers for OHS training program”. However, in the last paragraph of the Introduction section, the authors also mentioned “The present study identify the formation mechanism, … present the design framework of OHS trainings and education, ….then we construct the decision model …”. It seems that these two are not consistent.
  5. According to the title, the study focused on the “perceptions of construction workers for OHS training program”. Nevertheless, the Introduction section does not include sufficient background information related to OHS perceptions of workers. Although the authors explained that the related explanations can be found in the literature review section, it is suggested that some related explanations should be put in the Introduction section given that the “perceptions of construction workers for OHS training program” is the core of this study.
  6. More explanations about the table 2 and figure 2 are provided, which is good. Nevertheless, these explanations are very limited and not specific. In addition, according to the newly added information in page 5, if the study used “several rounds of cycles by participants including decision making level … enterprises, managers, governors, law marker and experts …”, these should be one part of the research method. For instance, how these participants were recruited and how many of them were invited to take part in the study, and some other similar questions related to this?
  7. In the section “4. Results and Discussion”, there can be found research method related contents, such as the first paragraph of the section 4.1 and the first a few paragraphs of the section 4.2.
  8. Discussions about the results are still very limited in the section 4.
  9. It seems that there is no response to the reviewer’s previous comment “Both the theoretical and practical implications of this study should be more discussed.”

Reviewer 3 Report

The discussion can be improved as to how this research adds to previous studies that have been discussed in the literature review section.